# Molecular physiology of Antarctic diatom natural assemblages and bloom event reveal insights into strategies contributing to their ecological success

Carly M. Moreno,[1] Margaret Bernish,[1] Meredith G. Meyer,[1] Zuchuan Li,[2] Nicole Waite,[3] Natalie R. Cohen,[4] Oscar Schofield,[3] Adrian Marchetti[1]

**ABSTRACT**  The continental shelf of the Western Antarctic Peninsula (WAP) is a highly variable system characterized by strong cross-shelf gradients, rapid regional change, and large blooms of phytoplankton, notably diatoms. Rapid environmental changes coincide with shifts in plankton community composition and productivity, food web dynamics, and biogeochemistry. Despite the progress in identifying important environmental factors influencing plankton community composition in the WAP, the molecular basis for their survival in this oceanic region, as well as variations in species abundance, metabolism, and distribution, remains largely unresolved. Across a gradient of physicochemical parameters, we analyzed the metabolic profiles of phytoplankton as assessed through metatranscriptomic sequencing. Distinct phytoplankton communities and metabolisms closely mirrored the strong gradients in oceanographic parameters that existed from coastal to offshore regions. Diatoms were abundant in coastal, southern regions, where colder and fresher waters were conducive to a bloom of the centric diatom, *Actinocyclus*. Members of this genus invested heavily in growth and energy production; carbohydrate, amino acid, and nucleotide biosynthesis pathways; and coping with oxidative stress, resulting in uniquely expressed metabolic profiles compared to other diatoms. We observed strong molecular evidence for iron limitation in shelf and slope regions of the WAP, where diatoms in these regions employed iron-starvation induced proteins, a geranylgeranyl reductase, aquaporins, and urease, among other strategies, while limiting the use of iron-containing proteins. The metatranscriptomic survey performed here reveals functional differences in diatom communities and provides further insight into the environmental factors influencing the growth of diatoms and their predicted response to changes in ocean conditions.

**IMPORTANCE**  In the Southern Ocean, phytoplankton must cope with harsh environmental conditions such as low light and growth-limiting concentrations of the micronutrient iron. Using metratranscriptomics, we assessed the influence of oceanographic variables on the diversity of the phytoplankton community composition and on the metabolic strategies of diatoms along the Western Antarctic Peninsula, a region undergoing rapid climate change. We found that cross-shelf differences in oceanographic parameters such as temperature and variable nutrient concentrations account for most of the differences in phytoplankton community composition and metabolism. We opportunistically characterized the metabolic underpinnings of a large bloom of the centric diatom *Actinocyclus* in coastal waters of the WAP. Our results indicate that physicochemical differences from onshore to offshore are stronger than between southern and northern regions of the WAP; however, these trends could change in the future, resulting in poleward shifts in functional differences in diatom communities and phytoplankton blooms.

Address correspondence to Adrian Marchetti, amarchetti@unc.edu.

The authors declare no conflict of interest.

See the funding table on p. 17.

KEYWORDS   polar diatoms, microbial ecology, metatranscriptome, Western Antarctic Peninsula, oceanography

The Western Antarctic Peninsula (WAP) is surrounded by a highly productive marine environment where phytoplankton such as large diatoms form the base of a rich polar marine food web (1–3). Diatoms are one of the main contributors to large phytoplankton blooms and carbon fluxes that occur along the coast and sea ice edge of the WAP (4, 5). Certain diatoms are efficient vectors of carbon export to the deep ocean due to their large size and heavily silicified frustules, along with efficient grazing by their euphausiid predators (6–8). However, the WAP has been undergoing rapid climate change with substantial wintertime atmospheric warming since the 1950s (9), drastic decreases in the duration of seasonal sea-ice coverage (10), and accelerated retreat and melting of glaciers (11).

Over 28 years of sampling as part of the Palmer Long Term Ecological Research (Pal-LTER) Project has revealed factors that are critical in explaining phytoplankton dynamics and triggering of blooms in the WAP (5, 12, 13). These include strengthening of winds over the Southern Ocean, an increasing positive trend in Southern Annular Mode, and ocean warming from incursions of warm circumpolar deep water onto the shelf (9, 14), all of which control upper ocean stability, thus influencing nutrient and light availability to phytoplankton.

Although the WAP is a highly variable system and warming has plateaued in recent years (14), regional warming persists, resulting in a latitudinal climate gradient in which a maritime subpolar climate in the north transitions to a dry polar climate in the south, differentially affecting phytoplankton community composition and productivity (15). In the northern region of the WAP, declines in sea ice and increased winds have resulted in deeper mixed layers and decreases in mean light levels in the upper water column (12), causing a significant reduction in summertime surface chlorophyll concentrations and more frequent occurrences of cryptophytes dominating the phytoplankton community (3). While cryptophytes have been detected throughout the entire WAP region (16), their tolerance for lower salinities (17) means they are more often associated with meltwater and do not typically co-occur with diatoms (18). The shift from diatom biomass to small cryptophytes could favor a transition in zooplankton grazers to salps because they are more efficient than krill at grazing small cells (3, 17), potentially resulting in major shifts in the distribution of krill and the higher trophic levels they sustain. In the southern region of the WAP, there is an increase in surface chlorophyll concentrations due to the reduction in permanently ice-covered regions, allowing more light to reach phytoplankton residing under the ice and fostering blooms of large centric diatoms (2), potentially resulting in long-term shifts in the phytoplankton community (19).

Large regions of the pelagic Southern Ocean are iron (Fe) limited. Natural and artificial Fe fertilization experiments in the region have shown that resident diatoms are Fe limited and that both pennate diatoms, such as *Fragilariopsis*, and centric diatoms, such as *Chaetoceros* and *Eucampia* (20, 21), are main responders to Fe inputs. Despite plentiful micro- and macronutrients in the coastal waters of the WAP, there exists a cross-shelf gradient in Fe concentrations with a possibility of transient Fe limitation offshore, where Fe-laden glacial meltwater has less influence (5, 22, 23), thus creating an Fe mosaic. In these regions, phytoplankton are believed to have evolved specific molecular mechanisms to cope with low Fe availability (24).

Numerous studies along the WAP have aided our understanding of the effects of changing environmental factors on phytoplankton community composition and structure through the use of high-performance liquid chromatography (HPLC) (25) and the 18S rRNA gene marker (6, 26) (e.g., see supplemental Text S1). In contrast, only a few studies have implemented gene expression analyses to investigate how polar diatom communities are able to subsist and thrive in the Southern Ocean. Metatranscriptome studies in the Ross Sea have demonstrated the potential for co-limitation between Fe and cobalamin (vitamin $B_{12}$) (27) and the interactive effects of Fe and ocean warming

on the metabolism of *Fragilariopsis* and *Pseudo-nitzschia* (28). In the northern Bransfeld Strait (north of Palmer Station and the LTER sampling grid), metatranscriptome analysis demonstrated that light-limited, sea-ice communities expressed decreased carbohydrate and energy metabolism and increased energy dissipation, suggesting either irradiance stress or inorganic C limitation (29). These studies provide a detailed understanding of diatom metabolism in natural and nutrient-amended assemblages; however, the metabolic priorities and Fe-related gene expression patterns of diatoms throughout the region, including those that exist along cross-shelf gradients, remain largely unresolved.

To better understand how environmental parameters such as Fe availability in the WAP influence the community structure and metabolism of phytoplankton, particularly that of diatoms, a metatranscriptome sequencing analysis was performed along the Pal-LTER sampling grid during the 2018 austral summer season. We identified oceanographic variables that influence eukaryotic phytoplankton composition in the surface waters of the WAP region. In addition, we examined the metabolic profiles of diatoms with an emphasis on expression patterns of genes involved in nutrient acquisition, photosynthesis, and Fe homeostasis, providing evidence of alternative metabolic strategies among ecologically important diatoms to cope with strong gradients in environmental conditions along the WAP. In addition, we opportunistically sampled a large *Actinocyclus* bloom in southern coastal sites of the Pal-LTER during our sampling period and provide a unique examination of the bloom through gene expression analysis. By leveraging recently sequenced transcriptomes from ecologically relevant polar phytoplankton species [e.g., see reference (24)], we sought to characterize the diversity and taxon-specific metabolic activities of WAP diatoms.

## RESULTS AND DISCUSSION

### Oceanographic environment across the Western Antarctic Peninsula region

During the 2018 Pal-LTER cruise, sea surface temperatures generally decreased north to south along the WAP, from 2.25°C to −1.77°C. Salinity gradually decreased from 33.77 PSU in northern shelf/slope stations to 33.01 PSU in southern, coastal stations. Fresher waters in coastal regions relative to those over the shelf or slope reflect the effects of sea ice meltwaters on water column structure (Fig. S1A; Table S1). The northern coastal region had warmer and saline waters, indicating this region was less influenced by glacial or sea ice melt and rather likely influenced by warmer upper circumpolar deep water (UCDW).

Historically, oceanographic conditions in the WAP have high spatiotemporal variability, although strong correlations between parameters are still observed (15, 25). Across the entire region, sea surface temperature and salinity were significantly positively correlated, along with dissolved inorganic carbon (DIC) and nitrate/nitrite concentrations. Chlorophyll *a* (Chl *a*), a measure of phytoplankton biomass, was significantly correlated with both primary productivity (PP) and bacterial productivity (BP); however, the two productivity rates were not significantly correlated to each other (Fig. 1B). Additional information pertaining to the oceanographic conditions associated with our observation period is provided in supplemental Text S2.

As many of the physicochemical parameters measured were strongly correlated (either positively or negatively) with each other, we performed a principal component analysis (PCA) to examine the differences in oceanographic variables across the different sampling sites, categorized into either (i) coastal, slope, or shelf stations (Fig. 1C) or (ii) North, South, or Far South stations (Fig. S1B). The first dimension (cross-shelf variability) explained 51.7% of the variance among samples and is represented by the strong negative correlation between nutrient concentrations and phytoplankton biomass (i.e., Chl *a*) that existed along a coastal to offshore gradient. Although nutrient concentrations were lower in coastal regions due to nutrient utilization by phytoplankton, fueling increased primary productivity, at no station was nitrate (+nitrite) concentration deemed to be fully depleted, thus limiting to phytoplankton growth (>15 μmol/L) (e.g., Fig. S1A). All surface macronutrients (i.e., nitrate, silicic acid, and phosphate) are generally found to

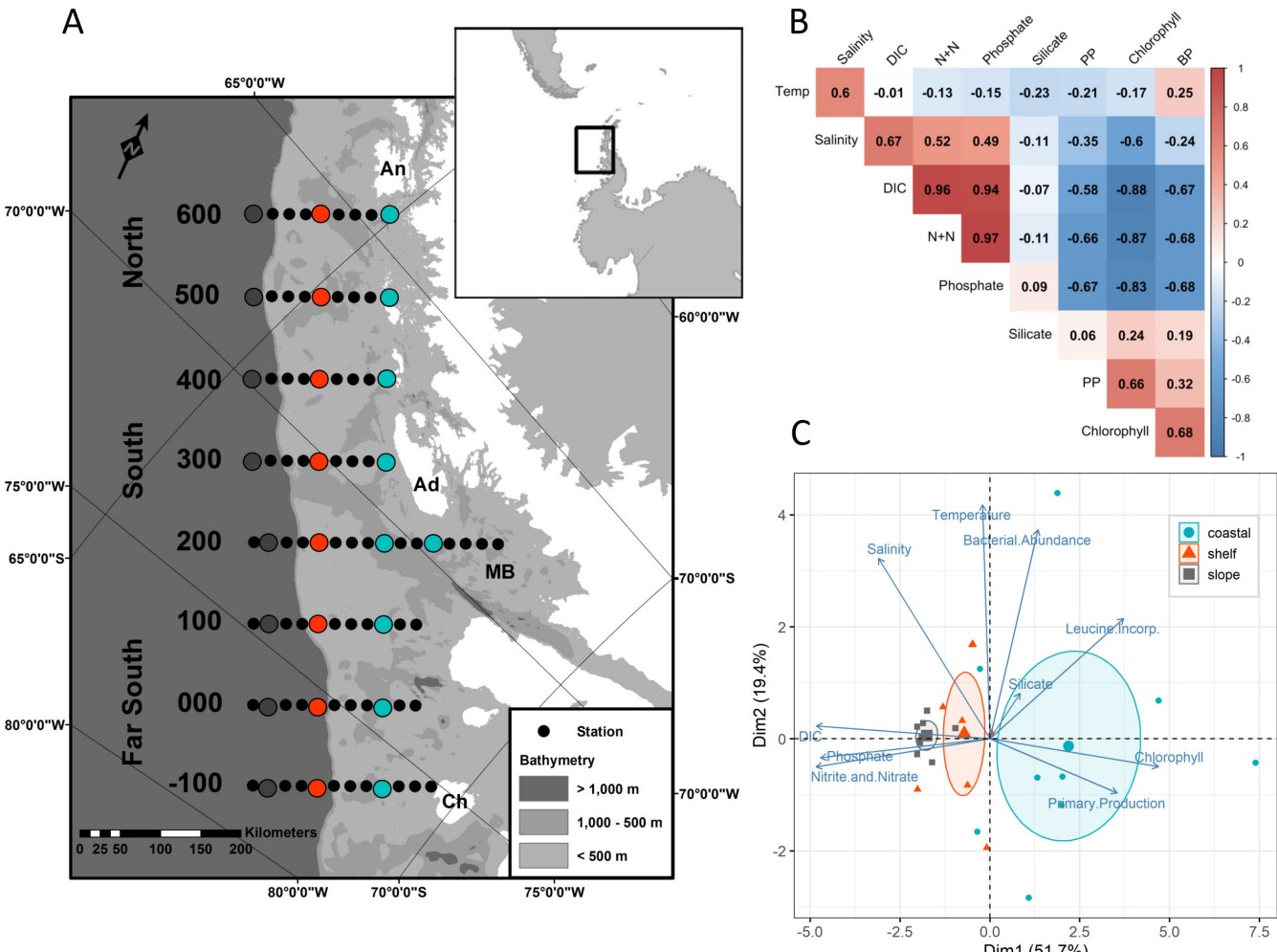

**FIG 1** (A) Location of the Palmer-LTER sampling grid along the Western Antarctic Peninsula. Adapted from Conroy et al. (30) with permission. Larger colored circles indicate stations where RNA was collected and are separated into regions based on legend in C. (B) Pearson's correlation values between environmental parameters with red colors indicating positive correlations and blue colors indicating negative correlations. Temp is temperature, DIC is dissolved inorganic carbon, N+N is nitrate and nitrite, PP is primary productivity, Chlorophyll is chlorophyll *a*, BP is bacterial productivity. (C) Principal component analysis bi-plot of environmental measurements in surface waters, with groupings of stations according to coastal, shelf, and slope regions. Ellipses represent the 95% CI around group mean points (the slightly larger point in the center of each ellipse).

be in relatively high concentrations in the Southern Ocean. This is primarily a result of there being high nutrient concentrations in the deep waters, deep winter mixing from the UCDW that resupplies the surface layer following biological depletion, and limitation of phytoplankton growth by other resources (e.g., Fe and light) (22, 31). The second dimension (along-shelf variability) explained 19.4% of the variation among samples and was represented by oceanographic parameters that varied along a north to south gradient (Fig. S1B). Thus, in relation to factors influencing phytoplankton, the spatial cross-shelf differences were more significant than those present from north to south.

## Eukaryotic plankton community composition and diversity

Metatranscriptome assembly and taxonomic annotation of sequence read counts to assembled contigs revealed that over the entire WAP region, dinoflagellates (37.9%), diatoms (23.1%), haptophytes (21%), and cryptophytes (2.7%) constituted the dominant taxonomic plankton groups among the transcript pool (Fig. 2A; Fig. S2). Samples from the northern sites (line 400–600) were predominantly composed of transcripts assigned to dinoflagellates, while those from the far southern coastal stations (−100 to 100)

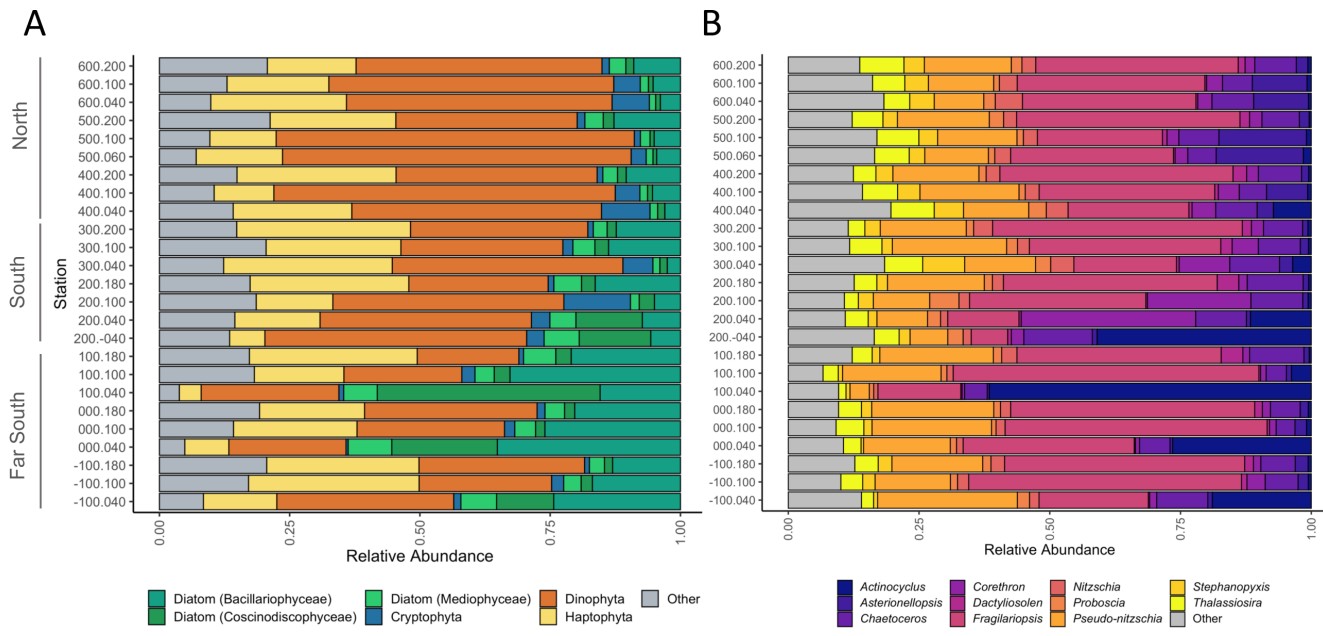

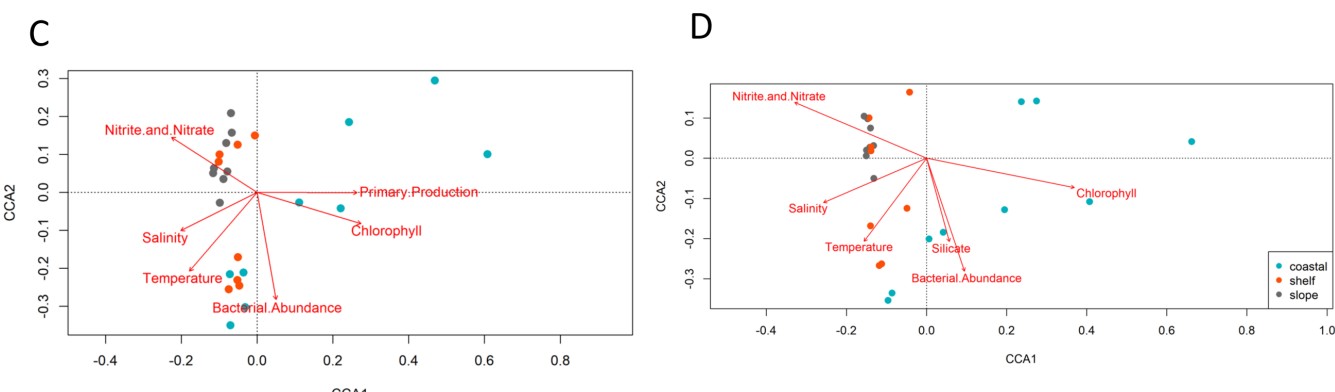

FIG 2 (A) Taxonomic proportion of RNA transcripts recruiting to the top most abundant phytoplankton groups. Diatoms are further delineated into pennate, radial, and bi-multipolar centrics. Gray indicates all other taxonomic annotations. (B) Taxonomic annotations of the top 10 most abundant diatom groups. Gray indicates all other diatom annotations. Canonical correspondence analysis (CCA) bi-plots of (C) the entire phytoplankton community and (D) the diatom community at each station are colored by distance from shore, wherein coastal stations are represented by turquoise, shelf stations are represented by orange, and slope stations are represented in gray. Environmental variables are indicated by red vectors.

had proportionally more diatom sequences, the majority of which were attributed to pennate diatoms (14.6%) rather than centric diatoms (4.7%). Northern stations were also characterized as having more diatoms in shelf water sites, while far south sites also contained more diatoms on the coast. Shannon-Weiner diversity of the eukaryotic plankton community was not significantly different for either onshore to offshore or North to Far South regions (Fig. S4A).

HPLC pigment analysis revealed some inconsistencies in phytoplankton taxonomic composition compared with RNA transcripts. In general, HPLC-based methods contained higher proportions of diatoms, whereas the metatranscriptomic-based approach contained higher proportions of dinoflagellates. Peridinin, a pigment marker for dinoflagellates, constituted only 2.5% of the total pigment measured across all the HPLC samples, whereas transcripts associated with dinoflagellates constituted 38%, on average. This may be due to the large genome sizes in dinoflagellates and their use of

regulating gene expression through post-transcriptional regulation strategies (Fig. S3). The diatom marker fucoxanthin constituted the highest relative proportion of pigment, 46% across all stations, and was at highest concentrations at coastal stations (e.g., 200.040 and 100.040). Additionally, alloxanthin, a marker for cryptophyta, was more abundant in northern stations and moving shoreward, whereas cryptophyte transcripts were generally rare across the study region.

Of the diatoms, transcripts assigned to the genus *Fragilariopsis* were numerically dominant over the majority of sites, especially those on the shelf and slope, reaching 52.5% of diatom transcripts at some sites. In contrast, *Actinocyclus*, a large centric diatom, constituted <1% of communities in several northern sites, yet in several Far South sites (stations 100.040 and 000.040 specifically) comprised up to 74% of all eukaryotic transcripts, associated with the transcriptome of an isolate of *Actinocyclus actinochilus* (UNC1410) previously sequenced from the WAP region. Additionally, station 100.040 had the highest primary production and Chl *a* measured during the cruise (Table S1), inferring a large bloom of *Actinocyclus* in the region (Fig. 2B).

Diatom diversity significantly increased northward, indicating that non-bloom stations were more diverse, i.e., not dominated by a single or few blooming genera/species (Fig. S4B). *Pseudo-nitzschia* constituted <32% of the diatom relative abundance at all stations, although up to 60% of *Pseudo-nitzschia* transcripts at some stations were assigned to *Pseudo-nitzschia subcurvata* at the species level, which was previously isolated from the WAP region and has a comprehensive transcriptome available (24). Other ecologically dominant centric diatoms such as those belonging to the genera *Thalassiosira*, *Chaetoceros*, *Corethron*, and *Proboscia* were detected but typically comprised <10% of the diatom transcripts.

Higher abundances of centric diatoms have been observed using several identification methods in the WAP region. Previous 18S rRNA gene sequences are indicative of large blooms of *Stellerima* and *Proboscia* populations in regions around Palmer Station (600 line) (8) and Marguerite Bay (200 line) (32). At Rothera Station (200 line), seasonal analyses of plankton community composition measured by HPLC (32) and light and scanning electron microscopy (33) documented that the dominance of certain phytoplankton groups consistently corresponded to certain phases of seasonal progression. The phytoplankton community transitioned through three phases: from high biomass of the haptophyte, *Phaeocystis* in early spring when wind-driven mixed layer depths (MLDs) were still deep with low overall light levels, to a pennate diatom community dominated by *Fragilariopsis* as MLDs shoaled and light levels increased, and finally to a centric diatom community dominated by *Proboscia* in late summer when increased meltwater promoted a well-stratified cold surface layer. During our collection period, the dominance of the large centric diatom *Actinocyclus* in the coastal southern regions suggests there may be a mid- to late summer bloom pattern associated with several different diatom genera (see further discussion below). However, transcript abundance may vary in relation to physiological status of the organisms and the ratio of RNA:DNA by taxon (34, 35).

Cryptophytes have also been shown to be relatively abundant, contributing up to 50% of Chl *a* concentrations in this region (16), yet cryptophyte-assigned reads were not readily detected in our sequence libraries, generally comprising <2% of overall transcript abundance. Cryptophytes may not be as transcriptionally active as diatoms or dinoflagellates, which could account for their low relative transcript abundances, or there may be differences in messenger RNA (mRNA) recovery or representative database sequences (28). Based on HPLC-derived alloxanthin (a diagnostic pigment for chryptophytes), cryptophytes were abundant in northern stations (Fig. S3), which may also be the result of niche exclusion by dinoflagellates and diatoms as these two groups generally grow faster and can inhabit a wider range of temperatures and salinities (25, 36). Notwithstanding clear niche differentiation between these three groups and high interannual variability of environmental parameters, taxonomic annotations derived

from metatranscriptomes and the inferred eukaryotic plankton composition patterns should be interpreted with caution.

Canonical correspondence analysis (CCA) was used to investigate how plankton community structure as inferred from mRNA sequence libraries correlated with specific oceanographic variables. Broadly, community eukaryotic plankton community composition based on metatranscriptomic read counts reflected the regions from which the samples originated (Fig. 2C). Coastal communities were associated with waters containing more Chl *a* and higher primary productivity. Slope and shelf communities clustered into two groups, one correlated with higher temperatures and salinities in the North, and the other correlated with higher nutrient concentrations. The physico-chemical variables included in the analysis explain approximately 60% of the variance in community composition (CCA1 and CCA2), [permutational multivariate analysis of variance (PERMANOVA), $P < 0.001$]. Although bacterial production was weakly significantly correlated ($P = 0.05$) with eukaryotic plankton community structure, it is likely not a determinant, as previously discussed.

Inferred diatom communities were further segregated according to distance from shore (Fig. 2D). The first canonical root, explaining most of the diatom variation (48.1%), revealed a notable separation between coastal and offshore stations. The strongest predictors were temperature and silicic acid concentrations ($P = 0.01$) followed by nitrate/nitrite concentrations ($P = 0.04$), with all physicochemical variables explaining 71% of variance in the diatom communities (PERMANOVA $P < 0.001$). Overall, temperature and Chl *a* are the primary variables explaining transcript-derived diatom community composition. Other studies in this region demonstrated similar findings using either 18S rRNA gene sequencing or HPLC. In a decadal study of phytoplankton pigments along the WAP, Schofield et al. (25) found the environmental factors that favored a shallower MLD resulted in larger blooms of diatoms compared to other taxa. In a 5-year study, Lin et al. (19) observed high relative abundance of centric diatoms in waters characterized by colder temperatures, reduced macronutrient concentrations, and shallow MLD, indicative of a strong influence of sea ice melt (18). Furthermore, they suggested that keystone diatom taxa, such as *Thalassiosira*, *Odontella*, *Porosira*, *Actinocyclus*, *Proboscia*, and *Chaetoceros*, were primarily responsible for high net community productivity (NCP) and associated C export potential from the mixed layer.

## Variations in inferred diatom metabolism across WAP communities

To gain a better understanding of diatom metabolism, we first examined broad pathway-level gene expression patterns across the WAP. Specifically, assembled contigs assigned to diatoms and annotated with Kyoto Encyclopedia of Genes and Genomes (KEGG) Orthology (KO) identifiers that were annotated to a higher gene family or pathway level within a KEGG module (MO) were examined (Fig. S5). Generally, diatom metabolic pathways were similar across the WAP as these pathways are fundamental to cellular functioning in diatoms. Highly expressed genes included those associated with ribosome, central carbohydrate metabolism, carbon fixation, ATP synthesis, and spliceosome KEGG pathways. Yet, there were significant spatial variations in pathway metabolisms likely reflecting the changing oceanographic conditions in the region. Diatoms in coastal stations in southern regions appeared to invest proportionally more in these aforementioned pathways compared to those from slope and shelf regions in the North. In particular, *Actinocyclus* and other diatoms at station 100.040 invested heavily in carbohydrate metabolism, cysteine and methionine metabolism, protein processing, and nucleotide biosynthesis metabolism (pyrimidine and purine). From the apparent high investment in genetic information processing, energy metabolism, and carbohydrate/lipid metabolism, we propose that these diatoms were experiencing high rates of translation and transcription, supported by C fixation and ATP synthesis to meet the high-energy demands of rapidly blooming cells. Further information on pathway-level analysis of diatom genes are provided in supplemental Text S3.

To further contextualize spatial gene expression of diatoms within the WAP region, a weighted gene co-expression network analysis (WGCNA) was used to group KOs into clusters or modules (MEs) of highly correlated genes based on patterns of expression (37). These groups of co-expressed genes were then correlated with the measured oceanographic parameters. Over 4,000 diatom genes were clustered into nine MEs (Fig. 3A), reflecting correlations with oceanographic parameters along a coastal to slope gradient (Fig. 1B and C). The red and blue MEs, and to a lesser extent the yellow ME, contained highly expressed genes positively correlated with Chl *a*, primary production, and phaeopigments (a degradation product of Chl *a*) and negatively correlated with temperature, salinity, DIC, and macronutrient (i.e., nitrate/nitrite and phosphate) concentrations. In contrast, the turquoise, green and brown MEs, correlated with these parameters in the opposite direction, with highly expressed genes at stations where high nutrient concentrations and lower Chl *a* concentrations were present. Specifically examining the MEs with the strongest correlations in relation to Chl *a*, genes in the red module were uniquely highly expressed at southern coastal stations associated with the bloom of *Actinocyclus*, indicating that these blooming diatoms had a unique metabolic profile as examined through gene expression (Fig. 3B). In contrast, genes clustered in the turquoise ME showed increased expression in shelf and slope stations and were positively correlated with high salinity and nutrient concentrations but showed opposing expression patterns in coastal stations, where they were negatively correlated with these oceanographic parameters, highlighting the clear effects of the cross-shelf gradient in nutrients and salinity on gene expression (Fig. 3B).

By performing an enrichment analysis, we determined distinct metabolic investments by diatoms in the red and turquoise MEs, the two modules which displayed the most significant correlations with the oceanographic parameters measured (Fig. 3C and D). The *Actinocyclus* bloom stations, represented by the green ME, were significantly enriched in the KEGG pathways: ribosome, carbon metabolism, C fixation, oxidative phosphorylation, and biosynthesis of secondary metabolites and amino acids. Similar patterns of enriched pathways have been observed in blooms of sea ice diatoms in the Wilkens Ice Shelf (30), dinoflagellate blooms in the Neuse River Estuary, NC (38), and Fe-induced diatom blooms in the NE Pacific Ocean (39), perhaps indicating a common response in blooming diatoms associated with growth and energy production. Although the cells at this station appear to have been blooming, the enrichment of glycine, serine, and threonine metabolisms suggest an effort to reduce photorespiration, or stress from reactive oxygen species (ROS), likely due to possible photoinhibition or perhaps the onset of bloom termination (32, 40).

In contrast, shelf and slope stations, represented by the turquoise ME, were enriched in KEGG pathways representing stress and regulation of transcription and translation: spliceosome, cell cycle, nucleotide excision repair, glycan biosynthesis, and DNA replication (Fig. 3D). The enrichment of glycan biosynthesis suggests increased formation of glycoprotein-rich extracellular matrices which may be important under freezing temperatures as has been observed in *Fragilariopsis cylindrus* (41). They are also part of a diverse family of transmembrane proteins that are widely implicated in allorecognition, including the establishment of symbiosis and microbial interactions (42). We speculate that increased expression of this pathway under nutrient limitation may foster interactions with beneficial bacteria or other microbes. In summary, these transcriptional patterns indicate blooming diatoms in southern coastal stations and potentially Fe-stressed diatom communities on the shelf and slope.

## Sea ice edge bloom of a large centric diatom

Analysis of remote sensing-derived surface Chl *a* concentration, cell densities, and gene expression revealed a bloom of the centric diatom *Actinocyclus* in the far south coastal regions of the Pal-LTER study area (Fig. 2B and 4). Cell densities of *Actinocyclus* in surface waters peaked at station 100.040, achieving $2.2 \times 10^4$ cells/L. It is hypothesized that very large centric diatoms such as *Actinocyclus* can preferentially bloom in the Southern

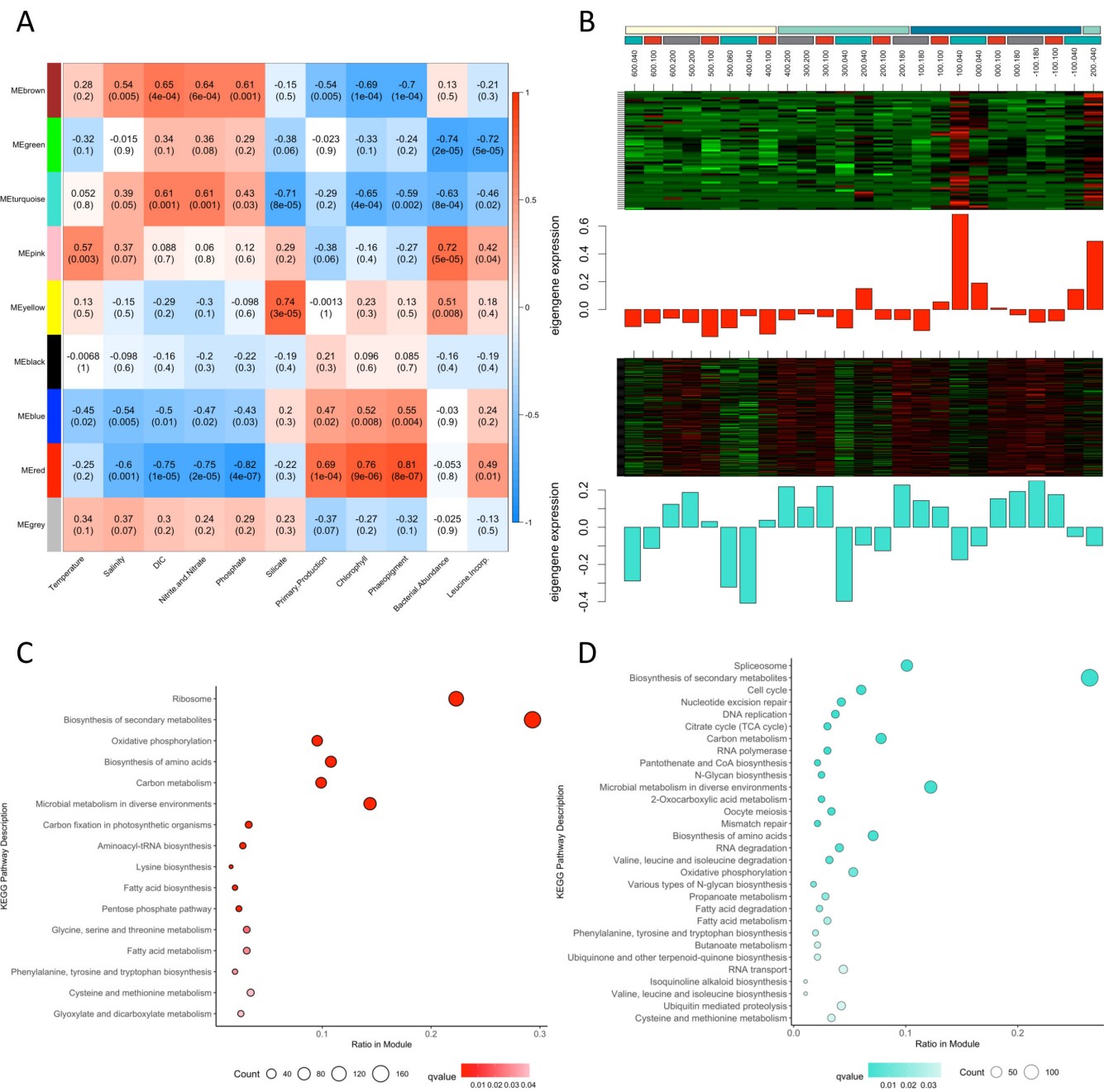

FIG 3 (A) Weighted gene co-expression network analysis demonstrates the correlation between environmental parameters (*x*-axis) and groups of similarly expressed genes assigned to modules (ME) and designated by arbitrary colors (*y*-axis). Top numbers in each cell are Pearson's correlation coefficients, with bottom numbers representing *P* values of the correlation test. The color of each cell indicates the correlation between ME and environmental parameters, with red indicating a positive correlation and blue indicating a negative correlation. Heatmap of transcripts per million gene expression plotted alongside bar graphs of eigengene expression within (B) the red module and (C) the turquoise module at each station. Dot plot of KEGG enrichment analysis of (C) the red ME and (D) the turquoise ME at the pathway level. Enrichment is based on the number of KOs overrepresented in KEGG pathways compared to all annotated KOs. Significance given to false discovery rate adjusted *P* values (*q* value) of <0.05. Darker color indicates higher significance. Counts represent the number of KOs detected in each pathway.

Ocean because their silica-laden frustules provide ample defense against grazers such as krill, whereas smaller diatoms cannot build up high biomass due to higher mortality as a result of top-down controls (7). Blooms of other large diatom genera (>20 µm) such as *Proboscia* and *Chaetoceros* have also been observed in pelagic and coastal regions of the Southern Ocean (20, 43, 44), particularly when Fe is replete and MLDs are shallow. Within

the WAP region, both *Proboscia* and *Actinocyclus* blooms have been previously documented in Marguerite Bay (45, 46), associated with cold, fresher surface layers that follow from sea ice retreat. Furthermore, larger blooms occur following years with more sea ice and delayed retreat (15). Eventually, a wind-induced breakdown of the meltwater layer causes a shift in the eukaryotic community from large blooming centric diatoms to smaller centric and pennate diatoms or cryptophytes (46).

Although *Actinocyclus* appears to be a frequent bloomer in the far south region, growth characteristics measured in laboratory isolates have demonstrated that members of this genus have relatively slower maximum growth rates compared to other polar centric and pennate diatoms (24, 47). *Actinocyclus* also appears to perform poorly under low iron conditions due to its high half-saturation constant for growth relative to iron concentration and is susceptible to photoinhibition at high light levels (47, 48). A transcriptome analysis of the *Actinocyclus* gene repertoire suggests that it may lack many genes encoding proteins involved in low-Fe coping strategies such as ferritin, an Fe-storage protein, or a complete high-affinity Fe uptake pathway (24). Thus, this diatom may be uniquely specialized for environments where cold meltwaters and associated elevated Fe levels are present, resulting in a boom-and-bust behavior that has been proposed as a lifestyle for some polar diatom species (44, 49) and is responsible for large amounts of C export (6).

Indeed, the *Actinocyclus* bloom we observed was characterized by the coldest and freshest waters measured along the WAP region in 2018 (Fig. S1). Satellite imagery of an 8-day composite during the sampling period demonstrated the high levels of Chl *a* in the region around station 100.040 (Fig. 4A), where maximum cell densities of *Actinocyclus* were observed among the bloom sites (Fig. 4B and C). The molecular investments made at the bloom stations were highest for ribosomal proteins, oxidative phosphorylation, and photosynthesis (Fig. S7 and S8). The region appeared to be naturally Fe replete as evidenced by the high expression of genes for Fe-requiring proteins, such as photosystem II subunits and cytochrome $b_6f$, and relatively low expression of iron starvation-induced proteins (*ISIP*s). This is consistent with previously measured cross-shelf gradients where coastal nearshore waters tend to have higher dissolved Fe concentrations compared to offshore waters (23).

Diatoms are known to have an increased ability to capitalize on newly available nutrient resources, with high initial investment in energy metabolism (50). It may be that while *Actinocyclus* exhibits slower growth rates and has more specialized niche requirements than diatoms such as *Fragilariopsis* or *Pseudo-nitzschia*, it can quickly allocate resources to growth. However, its large size, as demonstrated through images captured on an Imaging Flow Cytobot (Fig. 4C) and high Si requirement (27), indicates its main strategy is likely predator avoidance in a specialized niche. Interestingly, an *Actinocyclus* peroxiredoxin, an FK506-binding protein (*FKBP*), and a cold-shock protein (*CspA*) were highly expressed at non-bloom, shelf, and slope stations. FKBPs are a family of conserved proteins involved in diverse cellular functions including protein folding, cellular signaling, apoptosis, and transcription (51), whereas *CspA*s in bacteria such as *Escherichia coli* function as an RNA chaperone at low temperatures (52).

## Response to iron limitation by WAP diatoms

While light availability and degree of water column stratification are central to regulating productivity in coastal waters, micronutrients, especially Fe, have been suggested to be particularly critical in the Southern Ocean and offshore waters of the WAP region (4, 23, 53). Lines of evidence for Fe limitation in the WAP region are discussed in the supplemental Text S4.

To better understand how polar diatoms respond to the cross shelf gradient in Fe availability, we examined the expression of genes encoding Fe-responsive proteins such as those for Fe-dependent proteins and their functional replacements, as well as those proteins involved in ROS mitigation and nutrient acquisition. We focused on two dominant diatom genera, as measured through taxonomically annotated transcripts:

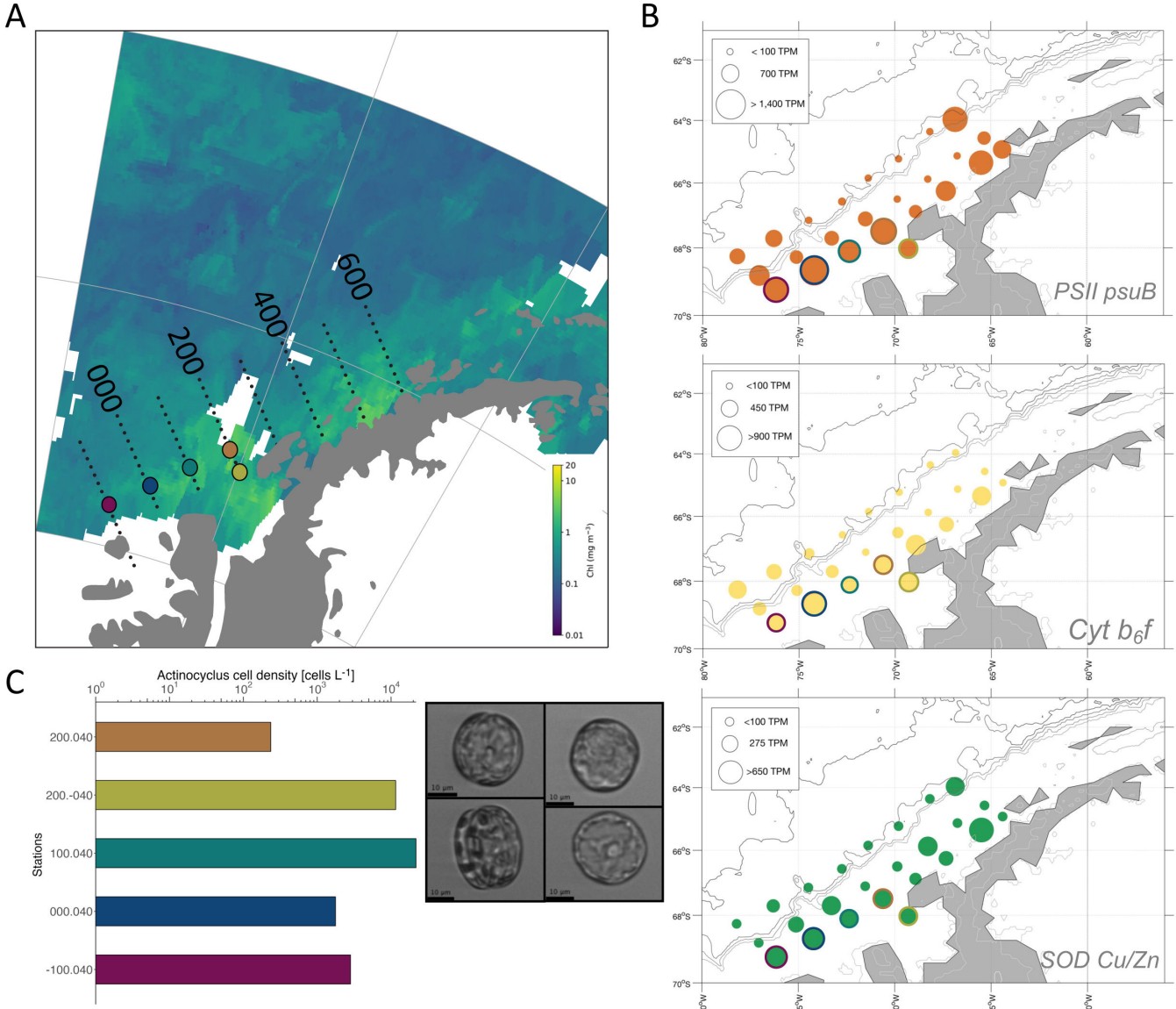

**FIG 4** (A) A large *Actinocyclus* sp. diatom bloom in the Far South coastal region of the Pal-LTER grid, visualized with an 8-day average composite chlorophyll *a* satellite image (9–16 January 2018). Larger, colored circles represent stations with high abundance of *Actinocyclus* sp. cells. (B) Log-transformed transcripts per million (TPM) of select *Actinocyclus* genes of interest of photosynthesis and reactive oxygen species that indicate actively growing diatoms. Stations outlined in colors represent the same bloom locations labeled on the satellite image. (C) Cell counts and images of *Actinocyclus* sp. captured via an Imaging FlowCytobot from surface seawater samples. Colored bar plots represent the same bloom locations labeled in panel A.

*Fragilariopsis* and *Pseudo-nitzschia*, both of which are also known to respond to Fe enrichment in high-nitrate, low-chlorophyll (HNLC) regions around the globe. Unlike *Actinocyclus*, which was primarily confined to coastal stations, members of these two diatom genera were detected with relatively high transcript abundances at all sampled stations (Fig. 2B).

Expression of Fe-related genes reflected an on/offshore gradient (Fig. 5A and B). Relatively high expression of the genes encoding the iron starvation induced protein 1 (*ISIP1*) suggests iron limitation occurs in shelf waters, regardless of latitudinal location along the WAP. ISIP1 is speculated to assist diatoms in taking up hydroxamate sidero-phores via endocytosis (33) and transcripts for the protein are more common in Southern Ocean diatoms compared to temperate diatoms (24). In *Fragilariopsis kerguelensis*, *ISIP1* was highly expressed under Fe limitation but was also expressed under low light

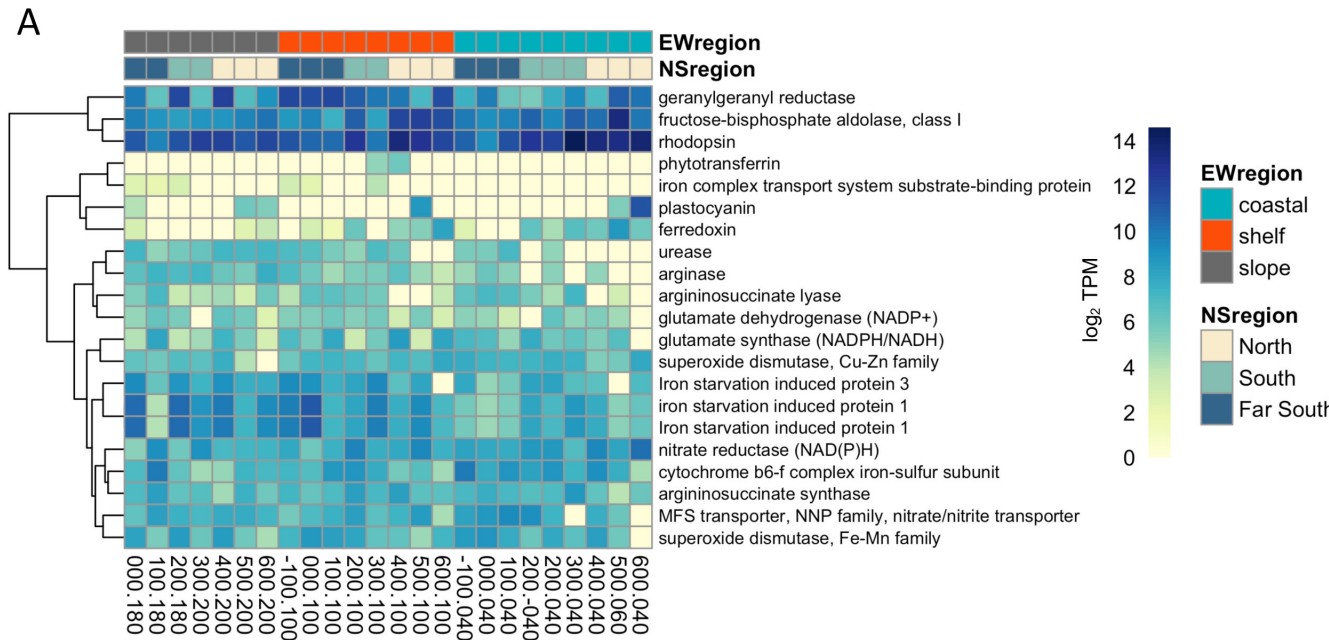

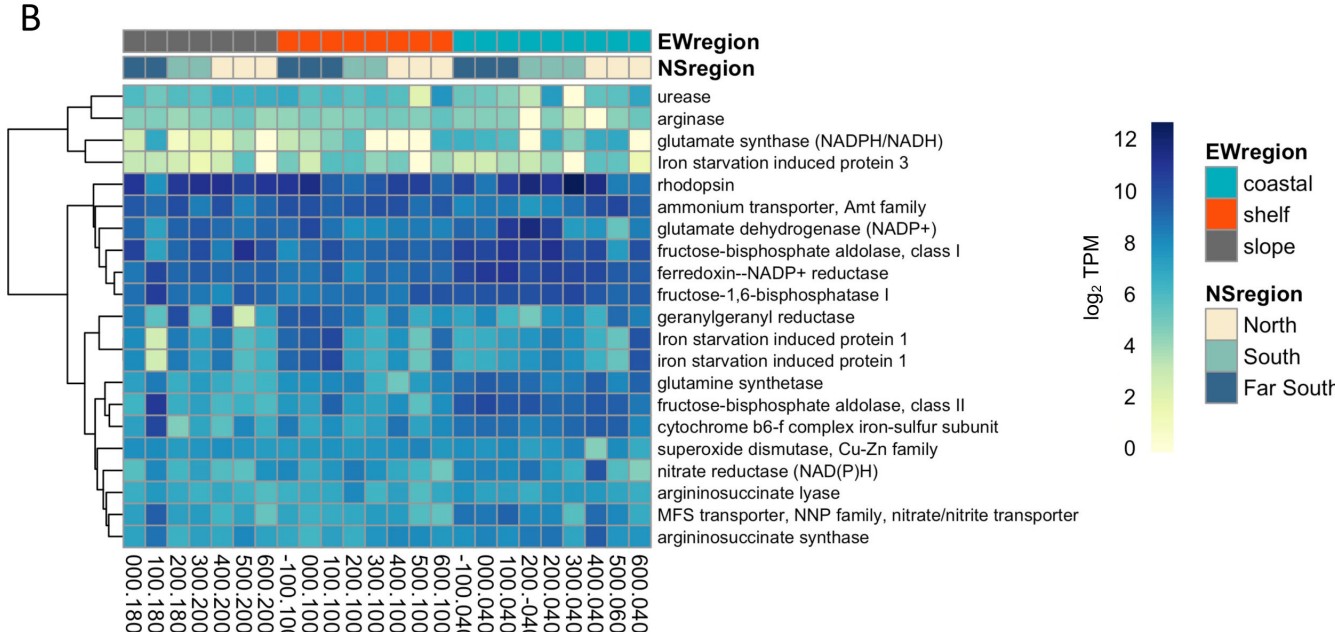

**FIG 5** Heatmap of RNA transcripts recruiting to KOs relating to N uptake, Fe homeostasis, and photosynthesis in the diatoms (A) *Pseudo-nitzschia* and (B) *Fragilariopsis*. Scale indicates transcripts per million (log base 2). Dendrograms show similarity in transcript abundances determined with Euclidean distances, and hierarchical clustering indicates transcripts.

conditions (54). This may explain its constitutive expression, even at low levels, across all stations in the WAP region. While phytotransferrin (previously *ISIP2a*) has also been proposed as a molecular indicator for Fe availability in polar diatoms (75), expression was only detected in *Pseudo-nitzschia* at a few stations. It may be that *ISIP1* could serve as a more useful molecular indicator of Fe limitation in this region.

Stations that had increased *ISIP1* expression also had higher abundances of a geranylgeranyl reductase (*ChlP*), a key precursor and regulator of Chl *a* synthesis and tocopherol production (40). *ChlP* may be useful in remodeling the photosynthetic

apparatus to cope with Fe limitation at offshore stations where Chl $a$ concentrations were reduced. As tocopherol is a strong antioxidant (40), *ChlP* is also likely involved in the removal of ROS that stems from the inefficient activity of an Fe-limited photosynthetic electron transport chain. Interestingly, genes encoding proton-pumping rhodopsins (*RHO*) were also highly expressed everywhere along the WAP region, but especially at coastal sites in the northern half of the grid, and often coincided with reduced *ISIP1* gene expression. *RHO* may be highly expressed in response to other environmental conditions other than low Fe concentrations in this region (55), for example, by high light levels in shallow MLDs of coastal stations (56). *Actinocyclus RHO* expression also followed a similar pattern (Fig. S9). Other photosynthesis Fe-requiring protein-encoding genes were highly expressed at coastal stations in both diatoms, including ferredoxin, cytochrome $b_6f$, and a vitamin $B_1$ synthesis gene, *THIC* (Fig. 5). Transcripts for the gene encoding plastocyanin, a protein used for Fe-independent transfer of electrons in photosynthesis, and a low Fe responsive aquaporin (*AqpZ*) (54) were also detected in the north, suggesting low Fe availability or other abiotic stressors. Slope stations expressed fewer transcripts compared to shelf or coastal regions.

The expression of genes encoding nutrient acquisition and metabolism were queried, along the WAP region. Nitrogen acquisition and assimilation require Fe and energy and should show contrasting patterns along a cross-shelf iron gradient. Both nitrate reductase and a nitrate/nitrite transporter (*NRT*) were more highly expressed in coastal waters mirroring the gene expression of *ISIP1* in shelf waters. The ornithine-urea cycle is important in recycling nitrogen throughout the cell (40), yet the genes we detected were not particularly abundant throughout the WAP region; the exception was urease, which had relatively higher expression in shelf and slope stations waters in both diatoms. Urease may be helpful in transporting urea or other reduced N compounds as it does not have an Fe requirement. Taken together, these molecular data suggest that diatoms in coastal regions are experiencing nutrient replete conditions with high irradiances, and within a relatively short distance from shore (~100 km) diatoms and other phytoplankton groups may be experiencing Fe limitation in an HNLC-like environment.

## Conclusions

Metatranscriptomes analyzed along the Pal-LTER survey elucidated the molecular physiology and metabolic flexibility of diatoms along the diverse habitats of the Western Antarctic Peninsula. Distinct phytoplankton communities and metabolisms were observed and closely mirrored the strong gradients in oceanographic parameters that existed from coastal to offshore regions. Diatoms were abundant in coastal regions especially in the Far South, where cold and fresher waters were conducive to a diatom bloom. Under these favorable conditions, the large centric diatom *Actinocyclus* was investing heavily in growth, energy production, and carbohydrate and amino acid and nucleotide biosynthesis pathways, resulting in a uniquely expressed metabolic profile in this Far South region.

In waters of the shelf and slope regions, we observed strong molecular physiological evidence of Fe limitation, supporting previous chemical and physiological studies. Diatoms in these regions employ *ISIP*s, a geranylgeranyl reductase, an aquaporin, and urease, among other strategies, while limiting the use of Fe-containing proteins, as inferred through gene expression analysis. The findings here reveal functional differences in diatom communities and is a first step in providing conclusive evidence for iron limitation and how certain ecologically dominant members of the phytoplankton community cope under these conditions.

The ecological success of diatoms along the WAP is likely related to their unique physiological adaptations, their resistance to a co-limitation of iron and light, and their unique molecular adaptations. The gene expression patterns observed here, although limited in providing only a snapshot in time, demonstrate the strong influence oceanographic factors may have in shaping disparate metabolic profiles of diatoms along the peninsula. This spatial analysis, along with additional time series analyses

of phytoplankton composition and diatom metabolism in this important region of the Southern Ocean, will contribute to our understanding of how these critical marine ecosystems might shift in future climate change scenarios, especially along a latitudinal gradient of the WAP, and will serve as a critical baseline for the understanding of the molecular physiology of natural plankton assemblages.

## MATERIALS AND METHODS

### Study location

The Pal-LTER study region is located along the western coast of the Antarctic Peninsula, encompassing an area of approximately 140,000 km$^2$ (Fig. 1A). Individual stations are arranged in a grid from the coast to offshore, running perpendicular to the peninsula, with major grid lines spaced 100 km apart and individual stations spaced 20 km apart along each line (15, 30). The region is divided further based on latitudinal, hydrographic, and sea ice conditions (57). The Northern region consists of lines 400–600; the South consists of lines 200 and 300, and the Far South consists of lines −100 to 100. Following an onshore to offshore gradient, stations 0.000–0.040 are within the coastal region; station 0.100 is within the shelf region; and stations 0.180 and 0.200 are over the continental slope. Samples were collected within a 2-week period during the Pal-LTER 2018 austral summer research cruise (30 December 2017–12 February 2018) aboard the ARSV *Laurence M. Gould*.

### Oceanographic data and sampling

All oceanographic data are publicly available through the Pal-LTER repository at https://portal.edirepository.org/nis/mapbrowse?packageid=knb-lter-pal.310.1, with detailed descriptions of sample collection and processing methods. In addition to sea surface temperature and salinity, Chl *a* and phytoplankton accessory pigment concentrations, PP, bacterial production and abundance, and inorganic nutrient concentrations were used in this analysis (Tables S1 and S2). Cell densities of the centric diatom *Actinocyclus* were estimated from discrete whole seawater samples passed through an Imaging FlowCytobot (58). For RNA, approximately 2–4 L of surface seawater was collected (depending on phytoplankton biomass) and filtered through a 47-mm 0.2-μm Supor filter (Millipore). Filtering was halted after 45 min, and each filter was preserved in 1 mL of RNA-later and stored at −80°C following a previous protocol (6). Samples were kept frozen on dry ice during shipment to the University of North Carolina at Chapel Hill. Prior to RNA extraction, filters were thawed and cut in half, with one half of the filter used for RNA sequencing analysis, while the other was archived.

### RNA extraction, sequence library preparation, and sequencing

Total RNA was extracted with the RNAqeuous 4PCR Kit (Ambion) according to manufacturer's instructions, with the addition of an initial 1-minute bead beating step using acid-washed sterile 425- to 600-μm glass beads (Sigma Aldrich) to ensure cells were mechanically removed and disrupted from the filter. Samples were eluted in 40 μL of sterile H$_2$O and stored in −80°C. Residual genomic DNA was removed by incubating RNA with deoxyribonuclease (DNase) I at 37°C for 45 minutes and purified by DNase I inactivation reagent (Life Technologies). Some samples required multiple incubations with DNase I. Sample concentrations and RNA integrity numbers (RINs) were determined using an Agilent Bioanalyzer 2100. RIN values were between 3.9 and 7.4. mRNA libraries were generated with ca. 2 μg of total RNA, using a poly-A selection primarily selecting mRNA of the eukaryotic plankton community, and prepared with the Illumina TruSeq mRNA Library Preparation Kit. Samples were individually barcoded and pooled prior to sequencing on a single lane of the Illumina HiSeq 4000 platform at Genewiz Sequencing Facility (S. Plainfield, NJ). Sequencing resulted in ca. 15 million 2 × 150 bp paired-end

reads per sample (Table S3). Sequences were submitted to the NCBI Sequence Read Archive under accession number PRJNA877830.

## Metatranscriptome assembly and annotation

Sequences were quality filtered with Trimmomatic v.0.38 and summary statistics were generated before quality filtering with FastQC v.0.11.8 and after Trimmomatic with MultiQC v.1.9. Trimmomatic (paired-end mode) removed Illumina adapters and used a 4-bp sliding window to remove quality scores below 20 and to keep sequences longer than 36 bp. Paired-end sequences from each sample that passed quality control were individually assembled using rnaSpades v.3.14.1, resulting in 25 individual assemblies of contigs. The resulting contigs from all samples were combined into one consensus assembly using cd-hit-est v.4.8.1 by clustering all samples at 100% identity, with a short word filter of size 11 (k-mer), and an alignment coverage of 98% for the shorter sequence (-aS). This consensus assembly was representative of all expressed genes of the microeukaryote community, along the WAP and was used in the subsequent annotation and read count steps.

Taxonomic annotations of consensus sequences were assigned using Diamond BLASTX v.2.0.4 based on homology (e-value cutoff of $10^{-5}$) to PhyloDB v.1.075, a custom database curated by the Allen Lab (Scripps Oceanographic Institute), consisting of eukaryotic genomes and the transcriptomes from the Marine Microbial Eukaryote Transcriptome Sequencing Project (59). We further supplemented the database with eight polar transcriptomes isolated from the WAP and previously sequenced (24). For this study, the individual polar transcriptomes were translated to protein space with GeneMarker-ES v.4.61 and functionally annotated with EggNOG-Mapper v.2.

For functional annotations, consensus sequences were searched against the KEGG database v.2018 by similarly using Diamond BLASTX and an $10^{-5}$ e-value. Annotation of proteins based on KEGG database classifications (https://github.com/ctberthiaume/keggannot) was performed using the python package Keggannot. Best hits to a functional protein and assigned KO identifiers and their higher-level classification, e.g., modules (MO) and Class 3 categories were obtained. Because KEGG only provides KOs for functionally annotated genes, KEGG gene definitions for certain targeted genes (e.g., *ISIP*s and rhodopsin sequences) were manually performed.

Trimmed reads from individual assemblies were aligned to the consensus assembly with Salmon v0.14.0 to obtain read counts. Read counts for consensus sequences that had taxonomic identifications from PhyloDB were used to evaluate plankton community composition of annotated transcripts. Read counts of consensus sequences that had an assigned KO identification number were used for functional annotations at the KEGG gene, module, and Class 3 level. Read counts of contigs sharing identical taxonomic and functional assignments were summed together. To account for shifts in community composition, reads were normalized within groups at the level of taxonomic interest (e.g., dinoflagellates, diatoms, and *Pseudo-nitzschia*) using transcripts per million (TPM).

## Metatranscriptome sequencing statistics

To examine the diversity and inferred metabolic activity of phytoplankton communities of the WAP region, 25 samples were collected from sites throughout the 2018 Pal-LTER cruise, extracted for RNA, poly-A selected, and were sequenced to a depth of 13–15 million paired-end reads per sample (Table S3). Following sequence quality control and assembly, over 15 million contigs were generated, with an N50 of 386, and combined into a large consensus assembly. Annotation of the consensus library for gene function (KEGG KO) and taxonomic association (PhyloDB) resulted in 2.3 million contigs with KEGG-annotations and 7.6 million contigs with taxonomic annotations. Of these, 38% of the KO annotations had an associated module (MO) level annotation, which links the assigned functional gene to a broader level KEGG category or pathway.

## Statistical analyses and data visualization

PCA was used with oceanographic data and specific genes of interest to visualize the spatial variability of samples along the WAP with the R package factoextra. To test if regionally grouped samples were statistically different, a PERMANOVA test was conducted. A CCA was used to determine the variance contribution of oceanographic factors on taxonomically annotated diatom RNA sequences in the R package vegan. Raw counts were log transformed to reduce weight given to taxa with small counts, and oceanographic variables that were strongly correlated together were removed from analysis, leaving the following to be included in the CCA: sea surface temperature, sea surface salinity, dissolved inorganic nitrogen (nitrate and nitrite), phosphate, silicic acid, chlorophyll *a* concentrations, primary production, and bacterial production. A PERMANOVA was used to determine the significance of the model. To visualize TPM gene expression at the KO, MO, and Kegg Class 3 levels, heatmaps were produced with the R package pheatmap and clustered using Euclidean distance and hierarchical clustering. Finally, to identify groups of highly correlated genes that co-occur across the WAP region and to quantify effects of oceanographic variables on those groups, we used a WGCNA (37). WGCNA clustered KOs and their associated $\log_2$ TPM-normalized counts into eigengene modules (ME, designated by arbitrary colors), representing the first principal component of the module matrix of similarly expressed genes. Using a minimum module size of 30 and dynamic tree cutting, 10 MEs emerged, 2 of which were combined into the "gray" ME. Finally, MEs were correlated with oceanographic and biochemical data by pairwise Pearson correlation coefficients and corresponding *P* values. A KEGG Class 3 enrichment test was performed with the R program clusterProfiler.

## ACKNOWLEDGMENTS

We are grateful to the crew of the ARSV Laurence M. Gould scientific support personnel, and Palmer LTER staff for their assistance in the field and lab. We thank Yajuan Lin and Sarah Davies for insight on sequence analysis and helpful comments on the manuscript.

This research was funded by the National Science Foundation Grants OPP1745036 (to A.M.) and PLR1440435 (to O.S.). C.M.M. was primarily supported by a Gates Millennium Fellowship.

## AUTHOR AFFILIATIONS

[1]Department of Earth, Marine and Environmental Sciences, University of North Carolina at Chapel Hill, Chapel Hill, North Carolina, USA
[2]Division of Natural and Applied Science, Duke Kunshan University, Suzhou, Jiangsu, China
[3]Department of Marine and Coastal Sciences, Rutgers University, New Brunswick, New Jersey, USA
[4]Skidaway Institute of Oceanography, University of Georgia, Savannah, Georgia, USA

## PRESENT ADDRESS

Carly M. Moreno, Marine Microbiomics Laboratory, Biology Program, New York University Abu Dhabi, Abu Dhabi, UAE

## AUTHOR ORCIDs

Carly M. Moreno  http://orcid.org/0000-0002-3046-1014
Adrian Marchetti  http://orcid.org/0000-0003-4608-4775

## FUNDING

| Funder | Grant(s) | Author(s) |
|---|---|---|
| National Science Foundation (NSF) | PLR1440435 | Oscar Schofield |
| National Science Foundation (NSF) | OPP1745036 | Adrian Marchetti |

## AUTHOR CONTRIBUTIONS

Carly M. Moreno, Conceptualization, Data curation, Formal analysis, Investigation, Methodology, Project administration, Resources, Validation, Visualization, Writing – original draft, Writing – review and editing | Margaret Bernish, Formal analysis, Investigation, Methodology, Writing – review and editing | Meredith G. Meyer, Formal analysis, Methodology, Software | Zuchuan Li, Formal analysis, Methodology, Software, Visualization | Natalie R. Cohen, Methodology, Software, Validation, Writing – review and editing | Oscar Schofield, Funding acquisition, Investigation, Methodology, Resources, Writing – review and editing | Adrian Marchetti, Conceptualization, Data curation, Formal analysis, Funding acquisition, Investigation, Methodology, Project administration, Resources, Supervision, Validation, Visualization, Writing – original draft, Writing – review and editing.

## DATA AVAILABILITY

Raw sequences are available at the National Center for Biotechnology Information Sequence Read Archive under accession number PRJNA877830. Processed sequence files are available at Zenodo under DOI 10.5281/zenodo.10620539. All Palmer Long Term Ecological Research data can be found at https://portal.edirepository.org/nis/home.jsp.

## ADDITIONAL FILES

The following material is available online.

### Supplemental Material

**Supplemental material (mSystems01306-23-s0001.docx).** Supplemental text, figures, and tables.

### Open Peer Review

**PEER REVIEW HISTORY (review-history.pdf).** An accounting of the reviewer comments and feedback.

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
