## [Reviewer comments · mSystems]

Molecular physiology of Antarctic diatom natural assemblages and bloom event reveal insights into strategies contributing to their ecological success

Carly Moreno, Maggie Bernish, Meredith Meyer, Zuchuan Li, Nicole Waite, Natalie Cohen, Oscar Schofield, and Adrian Marchetti

Corresponding Author(s): Adrian Marchetti, The University of North Carolina at Chapel Hill

Review Timeline:

Submission Date:	December 21, 2023
Editorial Decision:	January 4, 2024
Revision Received:	January 25, 2024
Accepted:	January 30, 2024

Editor: Hans Bernstein

Reviewer(s): The reviewers have opted to remain anonymous.

Transaction Report:

DOI: <https://doi.org/10.1128/msystems.01306-23>

Re: mSystems01306-23 (Molecular physiology of Antarctic diatom natural assemblages reveals multiple strategies contributing to their ecological success)

Dear Dr. Adrian Marchetti:

Overall, I think you (the authors) have done a nice job addressing the original reviewer's main concerns and think this is a good study that provides a rare look into the genome-encoded functions of diatoms and gene expression patterns from the WAP. The original reviewer was concerned about the limitations of how a transcriptomic "snapshot" can be used to infer larger ecological relationships. I agree with this concern but also with your response that this is indeed a good starting place for such inferences. However, I do encourage you to revise this paper to be more cognizant on the limitations and more clearly describe what is now known versus what is being inferred. Perhaps a revision of the title is also in order to describe that this is a short transect associated with a bloom event.

I agree with the original reviewer that the 1st paragraph (first couple of paragraphs, in my opinion) of the Results and Discussion reads more as if it belongs in the Introduction. My recommendation is to continue to reformat how this section leads off so that the reader is not directed to the supplementary information or overexposed to contextual information before being introduced to main results.

L209: section titled "Metatranscriptome sequencing statistics". Most of this text can be described in the Methods and thereby not stand in the way of your main Results and Discussion points.

L235: This additional HPLC analysis is helpful. Please add 1-2 sentences that critically evaluate how well the HPLC data support the relative abundances estimated from the metaT. Did the investigators do any qualitative taxonomic abundance measurements using light microscopy? This would also be useful if it existed.

L268: I think this requires some more conservative wording given the limitations of 18S rRNA-based taxonomy such as (for example) "...sequences indicative of *Stellerima* and *Proboscia* populations..."

Paragraph starting on L266: this is a very large paragraph after the revisions were added and it covers several topics. I recommend revising it into two paragraphs that (for example) break apart the topic of how the current transcript abundances are supported by known typical taxonomic abundances of diatoms and then the more detailed info on cryptophytes and possible niche exclusion. I would like to see at least some mention of how the current data supports the inference of niche exclusion as well.

L 355-373. Please avoid using the main text to describe how to interpret the figures. I recommend revising this section to focus more on the results and how the data supports them and use the figure legends to describe color coding assignments, etc.

L 443-444: "The region appeared to be naturally Fe replete..." Are there other supporting measurements for this? Please include and be specific.

L 460. Is it a profound result that the diatoms can be Fe limited in this region during blooms? Perhaps the subtitle and wording of the section below can be better focused on how the diatoms respond to Fe limitation. I am excited that this data set is well positioned to answerer this question from a functional level.

Revision Guidelines

- Upload point-by-point responses to the issues raised by the reviewers in a file named "Response to Reviewers," NOT IN YOUR COVER LETTER
- Upload a compare copy of the manuscript (without figures) as a "Marked-Up Manuscript" file
- Upload a clean .DOC/.DOCX version of the revised manuscript and remove the previous version
- Each figure must be uploaded as a separate, editable, high-resolution file (TIFF or EPS preferred), and any multipanel figures

must be assembled into one file

- Any supplemental material intended for posting by ASM should be uploaded separate from the main manuscript; you can combine all supplemental material into one file (preferred) or split it into a maximum of 10 files, with all associated legends included

Sincerely,
Hans Bernstein
Editor
mSystems

Dear Dr. Adrian Marchetti:

Overall, I think you (the authors) have done a nice job addressing the original reviewer's main concerns and think this is a good study that provides a rare look into the genome-encoded functions of diatoms and gene expression patterns from the WAP. The original reviewer was concerned about the limitations of how a transcriptomic "snapshot" can be used to infer larger ecological relationships. I agree with this concern but also with your response that this is indeed a good starting place for such inferences. However, I do encourage you to revise this paper to be more cognizant on the limitations and more clearly describe what is now known versus what is being inferred. Perhaps a revision of the title is also in order to describe that this is a short transect associated with a bloom event.

Author Response (AR): We thank the reviewer for taking the time to review our manuscript and their efforts to improve its clarity and focus. In accordance with the reviewer's suggestion, we have modified the manuscript title to the following:

‘Molecular physiology of Antarctic diatom natural assemblages and bloom event reveal insights into strategies contributing to their ecological success’

I agree with the original reviewer that the 1st paragraph (first couple of paragraphs, in my opinion) of the Results and Discussion reads more as if it belongs in the Introduction. My recommendation is to continue to reformat how this section leads off so that the reader is not directed to the supplementary information or overexposed to contextual information before being introduced to main results.

AR: In the revised manuscript, we have created a subsection in the Materials and Methods titled ‘Study Location’ and have moved the description of the Pal-LTER study region to this section. Line 574.

L209: section titled "Metatranscriptome sequencing statistics". Most of this text can be described in the Methods and thereby not stand in the way of your main Results and Discussion points.

AR: We moved this section to the Materials and Methods. Line 666.

L235: This additional HPLC analysis is helpful. Please add 1-2 sentences that critically evaluate how well the HPLC data support the relative abundances estimated from the metaT. Did the investigators do any qualitative taxonomic abundance measurements using light microscopy? This would also be useful if it existed.

AR: The paragraph has been revised to include the information requested from the reviewer.

Lines 236 – 250: HPLC pigment analysis revealed some inconsistencies in phytoplankton taxonomic composition compared with RNA transcripts. In general, HPLC-based methods contained higher proportions of diatoms, whereas the metatranscriptomic-based approach contained higher proportions of dinoflagellates. Peridinin, a pigment marker for dinoflagellates, only constituted ~6% of the total pigment measured across all the HPLC samples whereas transcripts associated with dinoflagellates, on average, constituted ~38%. This may be due to the large genome sizes in dinoflagellates and their use of regulating gene expression through post-transcriptional regulation strategies (Supplemental Fig. 3). The diatom marker fucoxanthin constituted the highest relative proportion of pigment, 46% across all stations, and was at highest concentrations at coastal stations (e.g., 200.-040 and 100.040). Additionally, alloxanthin, a marker for cryptophyta, was more abundant in northern stations and moving shoreward whereas cryptophyte transcripts were generally rare across the study region.

L268: I think this requires some more conservative wording given the limitations of 18S rRNA-based taxonomy such as (for example) "...sequences indicative of of *Stellerima* and *Proboscia* populations..."

AR: We revised this sentence.

Lines 276 – 278: Previous 18S rRNA gene sequences are indicative of large blooms of *Stellerima* and *Proboscia* populations in regions around Palmer Station (600 line) (8) and Marguerite Bay (200 line).

Paragraph starting on L266: this is a very large paragraph after the revisions were added and it covers several topics. I recommend revising it into two paragraphs that (for example) break apart the topic of how the current transcript abundances are supported by known typical taxonomic abundances of diatoms and then the more detailed info on cryptophytes and possible niche exclusion. I would like to see at least some mention of how the current data supports the inference of niche exclusion as well.

AR: As the reviewer suggested, we have divided this paragraph into two paragraphs. Our niche exclusion hypothesis is primarily supported by other studies referenced in this paragraph *ref. 25 and 35)

L 355-373. Please avoid using the main text to describe how to interpret the figures. I recommend revising this section to focus more on the results and how the data supports them and use the figure legends to describe color coding assignments, etc.

AR: We have revised this section as requested by the reviewer.

L 443-444: "The region appeared to be naturally Fe replete..." Are there other supporting measurements for this? Please include and be specific.

AR: Unfortunately, dissolved Fe concentrations were not measured during our observation period. However a previous multi-year study by Annett et al. 2017 has measured cross shelf gradients in dissolved Fe concentrations in this region. We have added the following sentence to the manuscript:

Line 457: This is consistent with previously measured cross-shelf gradients where coastal nearshore waters tend to have higher dissolved iron concentrations compared to offshore waters (23).

L 460. Is it a profound result that the diatoms can be Fe limited in this region during blooms? Perhaps the subtitle and wording of the section below can be better focused on how the diatoms respond to Fe limitation. I am excited that this data set is well positioned to answerer this question from a functional level.

AR: We changed the subtitle to: *Response to iron limitation by WAP diatoms*. As suggested by the reviewer, we also modified this section of the text to focus primarily on how diatoms respond to iron limitation. Line 470.

Re: mSystems01306-23R1 (Molecular physiology of Antarctic diatom natural assemblages and bloom event reveal insights into strategies contributing to their ecological success)

Dear Dr. Adrian Marchetti:

Your manuscript has been accepted, and I am forwarding it to the ASM production staff for publication. Your paper will first be checked to make sure all elements meet the technical requirements. ASM staff will contact you if anything needs to be revised before copyediting and production can begin. Otherwise, you will be notified when your proofs are ready to be viewed.

Featured Image Submissions: If you would like to submit a potential Featured Image, please email a file and a short legend to msystems@asmusa.org. Please note that we can only consider images that (i) the authors created or own and (ii) have not been previously published. By submitting, you agree that the image can be used under the same terms as the published article. Image File requirements: TIF/EPS, 7.5 inches wide by 8.25 inches tall (at least 2,250 pixels wide by 2,475 pixels tall), minimum 300 dpi resolution (600 dpi preferred), RGB, and no figure elements, e.g., arrows or panel labels. The legend should be a short description of the image, 1-2 sentences recommended.

We recognize that the video files can become quite large, so to avoid quality loss ASM suggests sending the video file via <https://www.wetransfer.com/>. When you have a final version of the video and the still ready to share, please send it to mSystems staff at msystems@asmusa.org.

Sincerely,
Hans Bernstein